# Host Protective Immunity against Severe Acute Respiratory Coronavirus 2 (SARS-CoV-2) and the COVID-19 Vaccine-Induced Immunity against SARS-CoV-2 and Its Variants

**DOI:** 10.3390/v14112541

**Published:** 2022-11-17

**Authors:** Rashed Noor

**Affiliations:** Department of Life Sciences (DLS), School of Environment and Life Sciences (SELS), Independent University, Bangladesh (IUB), Dhaka 1229, Bangladesh; rashednoor@iub.edu.bd; Tel.: +880-1749-401451

**Keywords:** COVID-19 pandemic, severe acute respiratory syndrome coronavirus 2 (SARS-CoV-2), variants of concern (VOC), COVID-19 vaccine-induced immunity, neutralizing antibodies (NAbs), immune memory

## Abstract

The world is now apparently at the last/recovery stage of the COVID-19 pandemic, starting from 29 December 2019, caused by the severe acute respiratory syndrome coronavirus 2 (SARS-CoV-2). With the progression of time, several mutations have taken place in the original SARS-CoV-2 Wuhan strain, which have generated variants of concern (VOC). Therefore, combatting COVID-19 has required the development of COVID-19 vaccines using several platforms. The immunity induced by those vaccines is vital to study in order to assure total protection against SARS-CoV-2 and its emerging variants. Indeed, understanding and identifying COVID-19 protection mechanisms or the host immune responses are of significance in terms of designing both new and repurposed drugs as well as the development of novel vaccines with few to no side effects. Detecting the immune mechanisms for host protection against SARS-CoV-2 and its variants is crucial for the development of novel COVID-19 vaccines as well as to monitor the effectiveness of the currently used vaccines worldwide. Immune memory in terms of the production of neutralizing antibodies (NAbs) during reinfection is also very crucial to formulate the vaccine administration schedule/vaccine doses. The response of antigen-specific antibodies and NAbs as well as T cell responses, along with the protective cytokine production and the innate immunity generated upon COVID-19 vaccination, are discussed in the current review in comparison to the features of naturally induced protective immunity.

## 1. Introduction

The severe acute respiratory syndrome γ coronavirus 2 (SARS-CoV-2) first evolved in Wuhan, in the Hubei Province of China in the latter part of December 2019, and from the middle of March 2020, the virus started to spread around the world, thus commencing the COVID-19 pandemic, causing 6,583,588 deaths out of 630,601,291 confirmed cases as of 10 November 2022 [1,2,3,4]. Thus far, many studies in the literature have discussed the SARS-CoV-2 genome, disease symptoms, transmission, pathogenesis, host protective immunity against the virus, the host immune evasion strategies by the SARS-CoV-2 wild-type strain and its variants, and the development of repurposed drugs and vaccines using various platforms to combat COVID-19 [5,6,7]. As discussed in previous studies, SARS-CoV-2 comprises a single-stranded, positive-sense RNA (+RNA) genome of ~29.9 kb, which encodes 16 nonstructural proteins (nsp1-16), 4 structural (S—spike, E—envelope, M—membrane, and N—nucleocapsid), and 6 accessory proteins (3a, 6, 7a, 7b, 8, and 9b), all of which are involved in the viral life cycle and pathogenesis as well as in the evasion of the host protective immunity [7,8]. Indeed, the SARS-CoV-2 genome exhibits significant genetic diversity, as noted from 7123 unique single nucleotide mutations/modifications among 12,754 complete US genome sequences as of 11 September 2020 [8,9]. As a result of such genetic shifts/alternations, the SARS-CoV-2 Wuhan strain has evolved into several variants of concern (VOCs), and the variants of interest (VOI) are Alpha (B.1.1.7), Beta (B.1.351), Gamma (P.1), Delta (B.1.617.2), and Omicron (B.1.1.529) [2,5,6,7,10]. To combat the infection caused by both the wild-type strain (the Wuhan strain) and its variants, a range of studies on host protective immunity have been conducted, various designs for new/repurposed drugs have been proposed, and COVID-19 vaccines using various platforms have been developed. In addition, many repurposed drugs and candidate vaccines are still being tested in several ongoing clinical trials [7,11,12].

The robustness of natural infection-induced immunity poses major consequences for reinfection and vaccine development [13]. Indeed, understanding immune induction, as well as immune memory, against viral infection and reinfection is of significance in terms of disease diagnosis and specific vaccine effectiveness [13,14]. Compared with the wild-type strain, the SARS-CoV-2 variants showed an increased frequency of infection as well as the evasion potential of the natural host protective immunity and, to some extent, the avoidance of vaccine-induced protection, thereby causing increased rates of infection and death [15,16]. A number of studies in the literature have described how the wild-type SARS-CoV-2 strain and the VOC strains may dodge the host protective immunity protection induced by both natural infection and by vaccination [15,17,18,19].

As of 8 November 2022, a total of 12,885,748,541 vaccine doses have been administered worldwide [1]. Most of the approved COVID-19 vaccines, including the BNT162b2 mRNA (BioNTech/Pfizer) vaccine, the mRNA-1273 (Moderna) vaccine, the ChAdOx1 nCoV-19 (University of Oxford/Astra-Zeneca) vaccines, the Gam-COVID-Vac (Gamaleya Research Institute) vaccine, the single-dose Ad26.COV2.S (Janssen) vaccine, the NVX-CoV2373 (Novavax) vaccine, the CoronaVac (Sinovac Biotech) vaccine, the BBIBP-CorV (Sinopharm) vaccine, and the BBV152 (Bharat Biotech) vaccines, have been designed to elicit both the cell-mediated (T cell responses along with the associated cytokine production) and humoral immune responses, in particular, the production of anti-S antibodies and neutralizing antibodies (NAbs) against the SARS-CoV-2 spike (S) protein [2,20,21,22,23,24]. As stated above, COVID-19 vaccines have been developed using the mRNA vaccine platform, the viral vaccine platform, the protein subunit vaccine platform, and the whole-cell inactivated virus vaccine platform, and they are currently in phase III trials around the world [20].

Indeed, a comprehensive knowledge of the immune response during COVID-19 vaccination is essential for the sake of global mass public health sustainability. More specifically, both the efficiency and efficacy of vaccines along with their related side effects, including possible hyperinflammation, need to be very carefully examined. Vaccine dosage, vaccine schedule, and the induction of antibodies upon vaccine administration should be estimated as well. Besides humoral immunity, the cellular immunity level upon vaccination also needs to be considered in order to understand the whole picture of vaccine-induced immunity. The key immunological data of the currently approved COVID-19 vaccines have already been discussed by several groups [13,20,22,25]. Many ongoing trials have been reported using a number of vaccines and exploring a wide range of populations with distinct demographic and genetic backgrounds. The immunogenicity and durability of the COVID-19 vaccines over natural infection-induced immunity would be interesting to investigate. Based on this foundation, the current review discusses the associated host protective immunity upon encountering SARS-CoV-2 and focuses on the COVID-19 vaccine-induced innate and adaptive immunity. The rationale for the second or booster doses of vaccines and the associated immunopathology are also discussed.

## 2. Induction of Host Immunity against SARS-CoV-2 Entry and Infection Stages

### 2.1. Viral Entry and the Initiation of SARS-CoV-2 Pathogenesis

SARS-CoV-2 pathogenesis starts during the entry of the virus into the host. As stated in earlier studies in the literature, viral entry is facilitated by the interaction between the spike (S) protein of the virus and the angiotensin-converting enzyme 2 (ACE2) receptor of the human host consisting of a metallopeptidase domain along with the HEXXH zinc-binding motif at the catalytic site [7,26,27]. After the S protein binds to the ACE2 receptor, the plasma-membrane-associated type-II transmembrane serine protease (TMPRSS2) instigates the S protein to accelerate membrane fusion, which is required for the release of the viral genome into the host cell [7,11,26]. The subunit S1 of the S protein consisting of the receptor-binding domain (RBD) binds to the peptidase (PD) domain of ACE2, and the S2 subunit mediates membrane fusion [28,29,30]. The S protein has a furin cleavage site at the S1–S2 boundary region, which, along with the three O-linked glycans, confers a high binding capacity to the ACE2 receptor. As stated above, after the S protein–ACE2 interaction, the virus employs TMPRSS2 for the activation of the S protein, which accelerates viral invasion [7,28,31]. Thus, SARS-CoV-2 infects the ACE2-expressing host alveolar epithelial type-II cells in the lungs, as shown in Figure 1.

#### 2.1.1. Immunological Consequences during Viral Entry and Hyperinflammation Effects

During viral entry, the innate immune signaling is activated through (1) the interaction between the pathogen-associated molecular patterns (PAMPs) of the virus (mainly the S protein and the viral genome) and the pattern recognition receptors (PRRs) or the viral RNA sensors, i.e., the toll-like receptors (TLRs), especially TLRs 3, 7, 8, and 9; the melanoma differentiation-associated protein 5 (MDA-5, a RIG-I-like receptor dsRNA helicase enzyme), and the RIG-I-like receptors (RLRs); and (2) the elevation of the messengers of the immune system, i.e., the accumulation of pro-inflammatory cytokines and chemokines, resulting in hyperinflammation or a cytokine storm [5,7]. Cytokine storm, or cytokine release syndrome (CRS), means a set of clinical conditions imparted by the accumulation of excessive immune cells and the immunological reactions, which, in turn, result in COVID-19 severity, i.e., the onset of the acute respiratory disease syndrome (ARDS), lymphopenia, etc., through systemic hyperinflammation [32,33]. The major pro-inflammatory cytokines include (1) interleukin (IL)-1, IL-2, IL-2 receptor-a (IL-2Ra), IL-6, IL-7, IL-8, IL-10, IL-15, IL-17, and IL-35; (2) tumor necrosis factor (TNF)-α; (3) granulocyte colony-stimulating factor (G-CSF); (4) interferon (IFN)-γ macrophage inflammatory protein-1α (MIP1α), MIP1β; (5) the monocyte chemoattractant protein-1 (MCP-1), which attract monocytes, neutrophils, professional phagocytes (dendritic cells, DCs), and non-professional phagocytes (macrophages), as well as T cells (both cytotoxic T cells (Tc) and T helper cells (Th, including Th1 and Th2)) to the site of infection, thereby promoting inflammation; and (6) interferons (IFNs), including IFN-α, and IFN-γ, which create the antiviral state [7,28,30,32,33]. Major chemokines include the IFN-γ-induced protein (IP 10 or CXCL 10), CCL-3, CCL-4, etc. [32] (Figure 1).

In the case of an impaired immune condition, the accumulation of such cytokines (especially with the impaired response of type-I IFNs in the early stage of COVID-19 infection) and chemokines and other immune cells, for instance, with the over-settlement of the resident macrophages in the lungs, may trigger hyperinflammation, which ultimately causes (1) damage to the lungs with ARDS and (2) the impairment of other major organs, including the kidneys, the liver, the spleen, the brain, the intestine, and the urinary tract [7,28,32,34]. Indeed, once the virus enters the body, it crosses through the bronchial tubes to the lungs and causes injury to the respiratory tree lining with the simultaneous irritation of the nerves of the airway lining, resulting in the inflammation and hardening of the mucous membranes, which causes the scarcity of oxygen supply to the blood, thus causing the shortness in breathing (ARDS), a state whereby the alveolar spaces are occupied by the neutrophils and monocytes, leading to hypercytokinemia [7].

### 2.2. Innate Immune Response Is Largely Mediated by the Interferons (IFNs)

The production of IFNs has been extensively discussed in previous studies, especially in studies by Aliyari et al. (2022) and Min et al. (2021) [5,35]. Generally, IFN-I and IFN-III create the cellular state of viral resistance in the SARS-CoV-2-infected host cells by means of autocrine and paracrine pathways, respectively [35]. As stated above, the IFN response (IFN-I and IFN-III) is the primary antiviral innate immune signaling pathway [5,36]. It should be noted that the alveolar macrophages (AMs), natural killer (NK) cells, DCs, and inflammatory monocyte–macrophages (IMMs) are the primary IFN producers [36]. While the products of interferon-stimulated genes (ISGs) impart the antiviral state, the IFN-I members (IFN-α and IFN-β) and the IFN-II member (IFN-γ) downregulate several host genes to hinder the viral infection [5]. The viral RNA, which is also detected in the endosomal compartments as well as on the cell surface, instigates the host IFN-I response (generating the antiviral state) through the membrane-bound TLRs, as mentioned earlier. The mechanism of the action of IFN-I mainly depends on its binding capacity to the IFN-α/IFN-β receptor (IFNAR) on the neighboring cells, along with the concomitant activation of the Janus kinase–signal transducer and activator of transcription (JAK–STAT) signaling pathway, which eventually increases the expression of ISGs [5].

#### 2.2.1. Molecular Mechanism of IFN Production in the Antiviral Innate Immunity

In the endosomes, TLR-3 detects the viral double-stranded (ds) RNA and activates the downstream adaptor protein Toll/IL-1 receptor (TIR) domain-containing adaptor (TRIF) [35,37]. The viral single-stranded (ss) RNA is recognized by TLR7, which instigates the downstream adaptor protein myeloid differentiation primary response gene 88 (MyD88) [5]. In the cytoplasm, the viral RNAs stimulate RIG-I and MDA5, which, in turn, induce the host mitochondrial antiviral signaling protein (MAVS), and such PRR-adaptor signaling cascades continue with the downstream kinases in order to activate transcription factors such as the IFN regulatory factor 3 (IRF3) and IRF7 [35]. The activation of the transcription factor nuclear factor-κB (NF-κB) is also imparted, mainly by the inhibitor of κB kinase (IKK)-α, IKK-β, and IKK-γ [35]. Eventually, the activated transcription factors translocate to the nucleus and enhance the expression of IFN-I, IFN-III, ISGs, and the pro-inflammatory cytokines. In addition, the cyclic GMP–AMP synthase (cGAS)-stimulator of the interferon gene (STING) pathway is also activated. The activated cGAS catalyzes the synthesis of 2′3′-cyclic GMP–AMP (2′3′-cGAMP), which subsequently binds to STING as the secondary messenger to activate TANK-binding kinase 1 (TBK1), which imparts IFN-I production, and other IKKs, thereby directing the downstream signaling cascades to trigger the expression of antiviral IFNs [35,38].

#### 2.2.2. Production of IFN-Stimulated Genes (ISGs) for Regulating Innate and Adaptive Immunity

IFNs bind to the IFN receptors on the cell surface and activate the Janus kinase 1 (JAK1) and tyrosine kinase 2 (TYK2), which then phosphorylate and instigate the signal transducer and activator of transcription proteins (STAT1 and STAT2), and ultimately combine with IRF9 to form the IFN-stimulated gene factor 3 (ISGF3) [35,36]. ISGF3 translocates into the nucleus and binds to the IFN-stimulated response elements (ISREs) to express ISGs (the signaling molecules or regulatory proteins of innate and adaptive immunity), which hinder different stages in the viral life cycle [35]. Indeed, the IFN response differs among COVID-19 patients, and it is obvious that an early IFN response is protective during the acute SARS-CoV-2 infection. Individuals bearing genetically disrupted IFN production or IFN signaling genes tend to be at high risk of COVID-19 severity [36].

#### 2.2.3. Evasion of IFN-Mediated Innate Immunity by SARS-CoV-2

A robust antiviral response is primarily dependent on the IFN response, and the mechanisms through which SARS-CoV-2 and its variants inhibit the IFN signaling pathways (IFN-I/IFN-III) along with the regulation of the ISG expression in favor of viral infection [10,36,39]. In order to confer an effective infection within the host cells, SARS-CoV-2 mounts several tactics to antagonize IFN signaling; for example, the IFN-β production is hindered by nsp1, 3, 5, and 12–15; ORF3a/b, 6, 7a/b, 8, and 9b; the N protein; and the M protein of the virus [39,40]. A detailed discussion of the evasion strategy can be found in many studies, particularly in studies by Znaidia et al. (2022), Kimura et al. (2021), and Beyer and Forero (2022) [10,39,40]. It should be noted that ORF6 hinders STAT1/2 nuclear translocation, which is a vital step to trigger the expression events of ISGs [40]. In addition, viral proteases such as nsp3 may also cleave the IFN-stimulated antiviral proteins (ISG15 as the corresponding example) [41].

## 3. Induction of Both Cell-Mediated Immunity and Humoral (Adaptive) Immunity

Both the cell-mediated immunity and humoral immunity are evoked during SARS-CoV-2 pathogenesis with the concomitant cytokine storm, as stated above [7,15,20,30,42,43]. After the neutrophils and monocytes/macrophages migrate to the site of infection, the increased level of pro-inflammatory cytokines (IL-6, IL-8, IL-12, TNF-α, etc., as shown in Figure 1) causes the immunopathology of COVID-19 in the lungs, i.e., the cytokine storm. At this stage, both the humoral and cellular immune responses are elicited. Indeed, both T cell and B cell responses against SARS-CoV-2 have been detected after a week of the COVID-19 symptoms. The CD8^+^ T cells can directly kill the virus-infected cells (cytotoxic activity), while the CD4^+^ T cells (especially Th1 cells) instigate both the CD8^+^ T cells and B cells and trigger the production of various cytokines to hurl the recruitment of the required immune cells [7].

### 3.1. Cell-Mediated Immunity

The infected epithelial cells may present viral PAMPS (RNA) to the CD8^+^ T cells, and along with the action of NK cells, those infected cells become cytotoxic and are killed by the antibody-dependent cellular cytotoxicity (ADCC) and apoptosis through NK-cell-secreted granules such as granzymes, perforins, etc. (Figure 1). Indeed, immune cells function within a complex regulatory network comprising T lymphocytes, B lymphocytes, monocytes, professional and nonprofessional phagocytes (i.e., DCs and macrophages, respectively), and the endothelial cells, which orchestrate the secretion of cytokines, principally IFN-γ, TNF-α, and IL-6 [33,44]. IL-6 binds to its soluble receptor to trigger monocytes’ differentiation into macrophages, which facilitates the recruitment of the immune components to the site of infection with concomitant inflammation [33]. IL-6 also inhibits the activities (as well as differentiation and development) of the regulatory T (Treg) cells and, thus, instigates acute immunopathological reactions [33,45].

Furthermore, T helper 17 (Th17) cells and T follicular helper (Tfh) cells differentiate into CD8^+^ cytotoxic T cells (CTLs), and B cells are activated [33]. According to Liu et al. (2021), leukocyte subpopulations seem to offer information about the severity of COVID-19 [46]. Eventually, the CD4^+^/CD8^+^ ratio (for Th and Tc cells, respectively), CD19^+^ (B cells), CD3^+^CD4^+^ (Th cells), CD3^+^CD8^+^ (suppressor T lymphocytes), and CD16^+56+^ (NK cells) may affect COVID-19 severity [46]. While the CD3^+^CD4^+^ and CD16^+56+^ lymphocyte counts are elevated in COVID-19 patients, TNF-α and IFN-γ decrease [46,47]. Moreover, Th cells (which help B cells secrete NAbs and modulate the immune response of other T cells) release IL-2, IL-4, IFN-γ, and other cytokines to stimulate macrophages and NK cells [46].

### 3.2. Humoral Immunity

As stated earlier, both the humoral and cellular immune responses are triggered by SARS-CoV-2 infection [14]. SARS-CoV-2-specific IgA and IgG remain along the mucosal sites and in the serum of infected individuals. IgM can be detected within 5–15 days of the onset of the COVID-19 symptoms, which peak within the first few weeks, followed by a decline after 2–3 months after infection [48]. IgA antibodies also rapidly diminish, while IgG antibodies seem to be more durable [48]. Along with the cellular cytotoxicity mediated by Tc, DCs present viral antigens to the CD4^+^T cells and induce their differentiation into memory Th1 and Th17 as well as memory T follicular helper (Tfh) effector CD4^+^T (Th2) cells (Figure 1). B cell responses simultaneously occur with Tfh cell responses, with a lag period of around one week after the onset of the symptoms [49]. The activated B cells differentiate, and the respective plasma cells produce anti-SARS-CoV-2 specific antibodies, i.e., IgM, IgA, and IgG [7,50].

As stated earlier, the initial inflammation caused by SARS-CoV-2 entry attracts the virus-specific T cells to the site of infection [7,30]. The antigen is then processed by antigen-presenting cells (APCs) and presented on the major histocompatibility complex II (MHCII) cleft whose recognition by T cell receptors (TCRs) activates the MHCII–TCR interaction in Th2 cells, which in turn activate the T cell–B cell interaction followed by antibody production by plasma cells, initially against the N protein and then against the S protein [7]. More specifically, IgMs are produced within a week, followed by IgG and IgA titers in 3–4 weeks [7]. It should be noted that acute COVID-19 patients possess more RBD-specific IgG antibodies than IgM and IgA [50]. The IgG1 and IgG3 subtypes are more predominant in the acute patient plasma samples after one week of the development of COVID-19 symptoms [50]. While the IgA antibodies decline after 3–4 weeks of symptom onset, IgG remains constant [51].

Concomitantly, both memory B cells and T cell subsets persist for a long time to combat reinfection. Regarding antibody exposure, another point is to ponder where the convalescent plasma (CP) therapy may be used, which is based on the passive immunity developed from SARS-CoV-2-recovered patients [52]. The recovered individuals possess high titers of the anti-SARS-CoV-2 NAbs in the serum, which can be introduced to severe COVID-19 patients, and the outcome is satisfactory [52,53].

## 4. Vaccine-Induced Protective Immunity

Vaccine-induced protection depends on the production of the different types of functional immunoglobulins (Igs), i.e., NAbs, against the SARS-CoV-2 spike (S) protein [20]. In general, vaccines are expected to generate NAbs in correspondence to their antigenic composition. In terms of immunogenicity, upon vaccination, both humoral and cellular immune responses are triggered (Figure 2). An adaptive immune response is conferred by B cells that produce antibodies through plasma cells (i.e., humoral immunity), while cellular immunity is imparted by the T cells [54,55]. Upon the injection of a vaccine into the muscle beneath the skin layers, the protein antigen is sequestered by professional phagocytes (DCs), which are activated through the interaction between danger-associated molecular patterns (DAMPs), i.e., the adjuvant used in the vaccine, and the PRRs attached to the DCs [21,56].

Danger signals facilitate the trafficking of the activated DCs to the draining lymph nodes, where the peptides of vaccine antigens are presented by MHC molecules on the DCs. Both MHCI and MHCII participate in the presentation of the antigenic peptide towards the T cells, in particular, MHCI to the CD4^+^T cells and MHCII to the CD8^+^T cells. The interaction is facilitated by the T cell receptors (TCRs) of the T cells. Thus, the DCs activate the corresponding T cells, which further interact with the B cell receptors (BCRs) of the B cells. Such T cell–B cell interaction instigates the development of B cells (i.e., B cell blast) in the lymph nodes, which results in the mellowing of the antibody response via the plasma cells (i.e., the generation of different antibody isotypes such as IgG, IgM, etc.) as well as in an elevated antibody affinity to the viral antigens [51]. The activated B cells instigate the production of short-lived plasma cells, which secrete the vaccine protein (antigen)-specific antibodies, and a rapid rise in the serum’s antibody titer over the next 2 weeks was observed [21,54,55].

Besides the generation of the effector B cells, the memory B cells are also generated for imparting immune memory. After the plasma cell differentiation, long-lived plasma cells, which continue to produce antibodies for years, migrate to the bone marrow niches [54]. Upon a natural infection by SARS-CoV-2, the CD8^+^ memory T cells quickly proliferate to encounter/kill the viruses, while the CD8^+^ effector T cells are engaged in the elimination of the virally infected cells, possibly with the aid of messenger molecules such as IL-12 and IFN-γ. NK cells are engaged in this process, together with ADCC. Indeed, the specific mechanisms of the actions of the currently used COVID-19 vaccines in correspondence to the SARS-CoV-2 life cycle in the host have been discussed in detail in the recent literature, whereby such a general mechanism of the action of vaccine can be noticed [2,21,24,57]. Moreover, one important clinical point to consider is that immune-mediated inflammatory diseases (IMIDs) are highly linked to COVID-19 severity and morbidity [15]. However, mRNA vaccines have been observed to induce both humoral and T-cell-mediated immunity [15,58].

An example of vaccine-induced immunity can be derived from the work of Huang et al. (2022), who analyzed the blood samples of healthcare workers who received vaccines and found an increased percentage of lymphocyte subpopulations from the peripheral blood of the individuals who received two doses over those receiving one dose [24]. Using in vitro simulation, they also observed that CD4^+^T cell subsets were elevated at a higher level at the post-vaccination stage than at the pre-vaccination state. TNF-γ, IFN-γ, IL-2, IL-17, IL-21, transforming growth (TGF)-β, and NK cells were also found to be increased. Finally, all samples were found with an increased level of anti-spike (S) antibodies (i.e., NAbs). An interesting finding was that memory Treg cells served as the autonomous predictors for those NAb levels [24].

### 4.1. Immunity Induced by the Major COVID-19 Vaccines

Overall, antigen-specific antibodies, NAbs, and T cell responses have been detected in the induced immunity by all COVID-19 vaccines. Regarding the longevity of the immunogenicity imparted by the major COVID-19 vaccines, Jamshidi et al. (2022) pointed to the acceptable immunogenicity of BNT162b1, BNT162b2, mRNA-1273, AZD1222, and the single-dose Ad26.COV2.S vaccines [22]. Sadarangani et al. (2021), as well as several other groups, thoroughly discussed COVID-19 vaccine-induced immunity [13,14,20]. The profiling of S-binding antibodies, NAbs, antibody avidity to the viral antigen, the accumulation of S-specific IgG^+^ memory B cells over time, and lastly, understanding the T cell immunity are of significance.

#### 4.1.1. mRNA Vaccines

The administration of the BNT162b2 mRNA (BioNTech/Pfizer) vaccine consisting of 30 μg mRNA (with two doses, 21 days apart) showed an increase in the CD4^+^ and CD8^+^T cells as well as increased antigen-specific production of IFNγ^+^ and IL-2 after the second dose [20,22]. The titer of the neutralizing S1-binding antibody was found to be significantly higher after the second dose than after the first dose [20,59,60]. mRNA-1273 (Moderna) consisting of 100 μg mRNA (with two doses, 28 days apart) showed substantial elevation in the CD4^+^ T cells secreting Th1-type cytokines, i.e., TNF-α, IL-2, and IFN-γ, while S-binding antibodies were detected 14 days after the administration of the first dose, which markedly increased after the second dose [20,26].

#### 4.1.2. Viral Vector Vaccines

The ChAdOx1 nCoV-19 (University of Oxford/Astra-Zeneca) vaccines consisting of 2.5–5 × 10^10^ viral particles, with two doses with a one-month interval, showed the highest T cell responses on the 14th day after the first dose and slightly higher than that after 28 days of the administration of the second dose; an increased amount of CD4^+^ T cells producing TNF-α and IFN-γ was also observed [61]. The S-binding antibody titer was detected within 14 days after the first dose, and the titer significantly heightened (with the isotypes IgG1 and IgG3) by 28 days after the second dose [62]. The titer of NAbs was also detected after the first dose, which increased by 14 days after the second dose [20,62].

The Gam-COVID-Vac (Gamaleya Research Institute) vaccine consisting of 10^11^ viral particles, with two doses with a 21-day interval, imparted CD4^+^ and CD8^+^ T cell responses, along with antigen-specific IFNγ secretion on the 14th day after the administration of the first dose [63]. As mentioned by Sadarangani et al. (2021), up to 89% S-binding antibodies and 61% NAbs were estimated over 14 days after the first dose, which significantly increased (up to 98% and 95%, respectively) after the second dose [20,64].

The recombinant, single-dose Ad26.COV2.S (Janssen) vaccine consisting of 5 × 10^10^ viral particles conferred CD4^+^ and CD8^+^ T cell responses with IFN-γ and/or IL-2 (suggestive of Th1 cell polarization) at 14 and 28 days of vaccination [20,64,65]. Both the S-binding antibodies and NAbs were estimated as sufficient within 84 days of vaccination [64,65]. Another single-dose, recombinant vaccine, Ad5-nCoV (CanSino Biologics), consisting of 5 × 10^10^ viral particles, imparted T cell responses with increased IFN-γ levels, as well as S-binding antibodies and NAbs over 28 days after vaccination [20,60].

#### 4.1.3. Protein Subunit Vaccines

The recombinant spike (S) protein nanoparticle NVX-CoV2373 (Novavax) vaccine (5 µg protein, two doses, 21 days apart), composed of the full-length recombinant spike protein trimers produced from the Wuhan-Hu-1 sequence followed by assembly into nanoparticles along with a saponin-based adjuvant (Matrix-M), was found to elicit both T cell and B cell immune responses against the S protein of SARS-CoV-2, like the other vaccines [66,67]. Cell-mediated immunity has been characterized by CD4^+^ T cell responses along with IFNγ, IL-2, and TNF-α production within 7 days after the second dose, and S-protein stimulation with the concomitant Th1 cell triggering was observed with the production of IL-5 and IL-13 [20].

The humoral response was observed through the titer of the S-binding antibodies detected 21 days after the first dose, which significantly increased after the second dose; the NAb titer was also found after the first administration, which vigorously increased by 7 days after the second dose [20]. The vaccine was found to be effective against SARS-CoV-2 B.1.351 in a phase IIb trial in South Africa and against B.1.1.7 in a phase III trial in the United Kingdom [66].

#### 4.1.4. Whole-Cell Inactivated Virus Vaccines

The CoronaVac (Sinovac Biotech) vaccine consisting of 3 µg protein, with two doses in a 14–28-day interval, showed specific CD4^+^T cells and the RBD-specific binding antibody as well as NAbs after 2 weeks, which increased after the second dose [31,68]. An interesting study was conducted by Zeng et al. (2022) to assess the immune persistence of the regular two-dose CoronaVac, which was compared with the immunogenicity of a third dose of the vaccine. The third dose administered 8 months after the second dose was found to elicit specific immune responses effectively (with high antibody titer), which had significantly declined 6 months after two doses [69]. The results, therefore, revealed that a two-dose schedule vaccination actually generated strong immune memory, and that is why the third dose given 2 months after the second dose triggered higher antibody titers than the initial two doses [69]. Schultz et al. (2022) also reported such immune robustness, whereby the booster dose was found to increase the NAb and T cells to recognize certain VOC strains including Delta and Omicron [68].

Like the other vaccines, the BBIBP-CorV (Sinopharm) vaccine, consisting of 4µg protein (two doses, 21 days apart) also showed an increased level of S-binding antibodies and NAbs after the second dose, compared with those observed after the first dose [70]. The study conducted by Gómez et al. (2022) to estimate the specific humoral response employing an Elecsys^®^ Anti-SARS-CoV-2 S assay and Ab neutralization cPass™ showed the highest antibody titer over 180 days after the second dose of BBBIBP-CorV, compared with the titers estimated after 90 days and 21 days [71,72].

Using the BBV152 (Bharat Biotech) vaccine consisting of 6µg protein (two doses, 28 days apart), Ella et al. (2021) conducted a double-blind, randomized, phase II clinical trial in healthy adults and adolescents (12–65 years) in India [20,73]. The assay of vaccine immunogenicity showed an increase in CD4^+^ CD45RO^+^ memory T cells on day 76 after the second dose, with the titer of anti-S-binding antibodies in 65% of the participants, which increased up to 98% by day 14 after the administration of the second dose [73]. NAbs were observed in 48% of the participants after the first dose and in 97% of the participants by day 14 after the second dose [73].

### 4.2. Effectiveness of COVID-19 Vaccines

The effectiveness of vaccines is certainly a prime concern for the sake of the complete treatment of COVID-19. A comparison of the running vaccines in trials in terms of their effectiveness is thus important. However, prior to discussing vaccine efficacy, it is worth noting that COVID-19 vaccines should be more examined for NAb production rather than their efficacy to hinder viral replication as part of innate immunity, which is mainly based on the production status of the required IFNs [74,75,76,77,78,79]. Since two doses of vaccines are recommended (the second dose is mainly to instigate memory B cell boosting for the production of NAbs in order to counter reinfection), it is relatively difficult to compare the efficacy of particular vaccines against a target population [21,54,68,71,72,74]. Although in some cases, the effects of vaccines just after the first dose are comparable, after the second dose administration, the results appear nearly the same for all vaccines [19]. Nevertheless, several groups are working on such a comparison of vaccine effectiveness, especially against SARS-CoV-2 variants. For example, the efficacy of the mRNA vaccine BNT162b2 and the viral vector vaccine ChAdOx1 nCoV-19 was compared by Lopez et al. (2021) against the SARS-CoV-2 Delta variant; both vaccines showed nearly similar efficacy after the second dose, while after the first dose, 36% efficacy was observed for the BNT162b2 vaccine and 30% efficacy for the ChAdOx1 nCoV-19 vaccine [19]. Although the effectiveness was measured to be 88% with two doses of the BNT162b2 vaccine and 67% for the ChAdOx1 nCoV-19 vaccine after the second dose, such a difference was considered to be small by Lopez et al. (2021) since the ChAdOx1 nCoV-19 vaccine ultimately showed effectiveness against the Delta variant [19]. Such a scenario has a positive impact on public health since the second dose administration at least ensures the sustainability of the immunogenicity of all vaccines beyond the so-called comparison among them [21,29,54,62,71,72,74,75].

#### 4.2.1. Perspectives on the COVID-19 Vaccine-Induced Durability of NAb Responses

The ultimate objective is to eliminate COVID-19 through ongoing approved vaccines, which are indeed in phase III trials and yet to receive a biological license. Therefore, drawing conclusions regarding both vaccine efficacy in a target population and the vaccines themselves (i.e., mRNA vaccines, viral vector vaccines, protein subunit vaccines, and whole-cell inactivated viral vaccines), which are still in phase III trials, is not clinically or statistically sound at this stage. Nevertheless, according to some studies and vaccine efficacy tests using small target populations, thus far, mRNA vaccines apparently seem to be highly effective against SARS-CoV-2 variants [22,24,25,26,59,72,73,74,75].

Another important point to consider regarding the second dose of COVID-19 vaccines is the durability of the circulating NAb responses against SARS-CoV-2 and its variants. Usually, natural infection imparts long-lasting NAb responses, i.e., persisting for up to a year in the cured individuals [76]. These NAb responses can be substantially boosted through immunization with another dose of BNT162b2 (BioNTech/Pfizer) vaccines and two doses of the CoronaVac (Sinovac) vaccine, as shown by Muena et al. (2021), although the particular span of the duration of NAb responses for the rest of the life has not been declared, nor is it possible in phase III trials, as stated earlier [68,76,77]. Nevertheless, in accordance with other reports, such a finding again suggests that a second dose or a booster dose of a vaccine instigates a significant elevation in B cell memory responses [21,54,68,71,72,76].

Finally, it is worth noting that the overall vaccine potency is also related to the initial triggering of innate immunity (armed with TLRs 3, 7, and 8; IFN-γ; TNF-α; APCs; IL-2, 6, 17, and 21; TGF-β; and NK cells and IFNs, as stated earlier), which is vital for the activation of protective humoral immunity/adaptive immunity including the elicitation of NAb production [24,33,34,46,47,78,79]. The stimulated immune cells of the innate immune system have been found to express elevated levels of co-stimulatory molecules, which in turn trigger T cells in the draining lymph nodes in order to activate B cells [78].

## 5. Conclusions

The present review discussed both the cellular and humoral immune response upon COVID-19 vaccination, which can be compared with the immunity that is elicited upon the natural SARS-CoV-2 infection. Indeed, elucidating the immune response of COVID-19 vaccination is important to provide consistent protection for individuals against SARS-CoV-2 and its variants. Concerns regarding the effectiveness of vaccines, the requirement of a second dose to elicit NAbs, the durability of COVID-19 vaccine-induced NAb responses, and the innate immune responses were addressed in the current review. Such studies/analyses may help to develop new approaches for immunization and vaccine development platforms in addition to the design of new drugs. Moreover, the immunological signal transduction pathways (including multiple immunological parameters) associated with the protection against SARS-CoV-2 and its variants need to be fully understood.

## Figures and Tables

**Figure 1 viruses-14-02541-f001:**
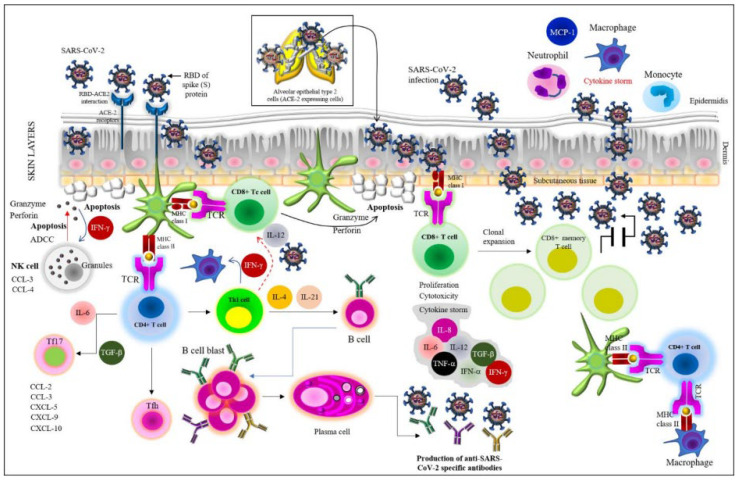
Induction of host immunity against SAR-CoV-2 infection. Details are given in the text.

**Figure 2 viruses-14-02541-f002:**
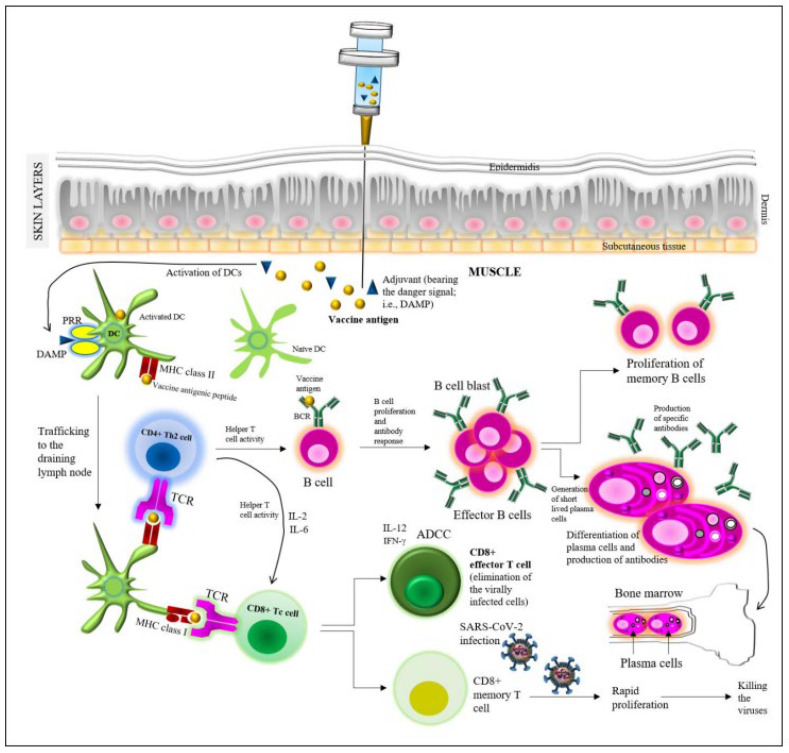
Vaccine-induced protection, adapted from Pollard and Bijker, 2021 [54]. Details are provided in the text.

## Data Availability

Not applicable.

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
