# Peer review of "Host Protective Immunity against Severe Acute Respiratory Coronavirus 2 (SARS-CoV-2) and the COVID-19 Vaccine-Induced Immunity against SARS-CoV-2 and Its Variants"

_viruses, 2022, doi:10.3390/v14112541_

Round 1

Reviewer 1 Report

The authors present a rather thorough review on the immune response to SARS-CoV2 as a result of natural infection and vaccination.  While the information presented is useful, the paper presents more like a literature review rather than a critical review of the literature.

For example, the authors describe the response of various cohorts to several commercial vaccines.  However, there was no attempt to compare the immune of each vaccine as to the ability to prevent infection with the virus or mitigate infection with the virus, also which vaccine(s) is/are most effective in generating an immune response?  Why does it require two vaccine doses in most cases to elicit a measurable NAb response?  How long do the NAb responses persist?  Do the vaccines induce a sufficient innate immune response to impact early viral replication within the host?

Some critical discussion on the above points and a critical comparison of the efficiency of each vaccine to induce both a cellular and humoral immune response would strengthen the paper.

Author Response

I would like to the thank the reviewer for raising several critical issues which are indeed mandatory to incorporate into the manuscript. I have addressed the reviewers’ query about the effectiveness of vaccines as well as the requirement of second dose to elicit NAb in Page 10: lines 426-449 (shown in red font). Two new references (reference serials 74 and 75) have been added to write this paragraph. This is to be noted that regarding the vaccine action in course of viral replication inhibition, a few lines have been added in lines 427-430 for which references 75-79 have been included since there is no discrete information about that although the innate immunity perspectives have been discussed in these papers.

     Regarding the durability of the COVID-19 vaccine induced NAb responses, I have added a few lines in Page 11: Lines 450-459 (shown in red font). Two new references (reference serials 76 and 77) have been added to write this paragraph.

     Regarding the innate immune generation by the vaccines, I have already stated that description in Page 6: lines 224-245; however, as per reviewer’s suggestion, I have added a few lines in Page 11: Lines 459-464 (shown in red font). A new reference (reference serial 78) has been added to write these couple of sentences. Besides, in the Abstract section, I have newly mentioned about the innate immunity as a part of discussion goal of this review (shown in red font).

Reviewer 2 Report

This reviewer has no major concerns about this review manuscript on Covid-19 and its current available vaccines. The whole structure and logic seem well-organized and the interpretations are clear and comprehensive. The same authors or group have already published many similar reviews on similar topics, which, to some extent, is an excellent effort. The reviewer would ask if the authors could add some perspectives as a final section of this review, with which the significance of this review would be better expressed with the unique point of view from the authors.

Author Response

Thank you so much for appreciating my work. As per reviewer’s comment, at the end of the revised manuscript (before the Conclusion section), I have added some critical perspectives regarding on the effectiveness of vaccines, requirement of second dose to elicit NAb, the durability of the COVID-19 vaccine induced NAb responses, and the innate immune generation by the vaccines in Page 10: lines 426-449, Page 11: Lines 450-464 (shown in red font). Five new references (reference serials 74-78) have been added to write this new section.

Reviewer 3 Report

This is an interesting review on the immunity against SARS-CoV-2 and the vaccines with a lot of information.

There are a lot of repetitions and a lot of information that is not well organized. All this information is given without a real comparison between the two kinds of immunity (caused by the virus and the vaccine), which would give a meaning to the review. There are significant errors (as an example the CD markers in l.239 are wrongly mentioned, the title 3 and many more).

According to the title the review should concern the protective immunity and the symptoms and the immunopathology of COVID,

Tables offering a summary of the data (references) are missing.

Author Response

I would like to thank the reviewer for the positive criticism and valuable suggestions. Regarding a real comparison between the two kinds of immunity (caused by the virus and the vaccine, I have already discretely written those in Sections 2 and 3 (the immunity induced by the viral infection) and Section 4 (the cellular and humoral immunity caused by the COVID-19 vaccines). The illustrations also show the clear difference between the immunity caused by the virus (Figure 1) and the vaccines (Figure 2). Indeed, showing these perspectives was the main goal of this review. Hence, at the end of the revised manuscript (before the Conclusion section), I have added two paragraphs explaining the effectiveness of vaccines, requirement of second dose to elicit NAb and their durability along with the induction of innate immunity induced by the vaccines in Page 10: lines 426-449, Page 11: Lines 450-464 (shown in red font). Several new references have been added (reference serials 74-78) to gather the information.

     Regarding the CD markers especially written within the lines 238-247, I rechecked but didn’t find mistakes. However, I tried to clarify a couple of terms for the ease of reading (as shown in red font). Yet, it would be highly appreciated if the reviewer kindly points the specific errors in the CD markers (if any).

     Regarding the immunopathology, lots have been already written (focusing on the cytokine storm/ hyper inflammation) in lines 115-149 and in lines 231-236 focusing on IL-6 action.

     Regarding the Table offering a summary of the data has not been used since Sadarangani et al. (2021) [reference serial 20] already used that sort of Table. I didn’t want to repeat the data/ write up; rather I discussed the points about the COVID-19 induced immunity in Section 4 with Figure 2, originally drawn by me. However, I’d be happy to do more if the reviewer suggests me further although I’m afraid that may be a repetition. Overall, the revisions I have made (slightly in the Abstract section, at the end of the Introduction section emphasizing on the objectives of the review; and the extensive revision in the last paragraphs before Conclusion section on the overall immunogenic perspectives) hopefully will satisfy the reviewer regarding the organization of the revised manuscript.

Round 2

Reviewer 1 Report

The authors have complied with the requests of the reviewer.  The added discussion on comparison of the vaccines is noteworthy and does add strength to the paper.  Would still like to see more critical review, but the authors have provided a reasonable and useful review on SARS Cov2 vaccines.

Author Response

RESPONSE TO REVIEWER 1:

I would like to the thank the reviewer for admitting the revised manuscript. Regarding the aspect of critical review, in round 1, I have addressed the concerns raised by the reviewer which included the effectiveness of vaccines, requirement of second dose to elicit NAb, durability of the COVID-19 vaccine induced NAb responses, and the innate immune response.

  1. Based on the reviewer’s comments in round 2, while correcting the manuscript, I found that in lines 82-88, I have written about the necessity of knowing the vaccine efficiency and efficacy with related side effects on which many other groups and I published several papers. Some other technical considerations like the vaccine dosage/ scheduling, induction of necessary cytokines and production of antibodies have also been included.
  2. Thinking about the reviewer’s valuable suggestion, I segregated the last portions into 4. and 4.2.1 subsections where the Perspectives are differently written. Moreover, along the whole manuscript, some important changes have been brought in the immunological perspectives (as shown in red font and the yellow highlights). I would be happy to revise more if the reviewer asks me specifically to do so.

Reviewer 3 Report

I would like to thank the author for his response. Unfortunately, the minor english remain, even simple mistakes like words repetitions (l 78). They are a lot and I cannot mention them one by one. The author should check the paper with a lot of attention.

Regarding the paragraph on lines 238-247 the sentence "the cell surface CD markers are important to measure the extend of COVID-19" should be changed to "leukocyte subpopulations seem to offer information about the severity of COVID-19". They are not the markers, but the populations determined by the markers that are affected. Then CD3+CD4+ cells are the Th cells and not the total T cell coreceptor (because they express both CD3 and CD4) CD16+CD56+ are not ADCC (ADCC antibody dependent cellular cytotoxicity is one of their functions and not the cells themselves).

l 266 plasma cells and not B cell blast produce antibodies.

l 268 IgA antibodies and not titer

l 272 IgG remains constant, not in the constant titer

l 345 not antigen-specific IFNγ but antigen-specific production of IFNγ. There is not antigen-specific IFNγ

I don't understand the meaning of the paragraph l 352-356

There are more of these mistakes. The author should check the whole paper

The added paragraph in l 425 should have a different title because it concerns all the vaccines. This paragraph needs a more careful check of the english, because it has a lot of mistakes.

Author Response

RESPONSE TO REVIEWER 3:

I would like to thank the reviewer for mentioning the specific corrections and the valuable suggestions. I have carefully gone through the whole manuscript; and tried to revise the manuscript according to the reviewer’s suggestions as appended below:

  1. I have corrected the word repetition in line 78.
  2. According the reviewer’s suggestion, I have gone through the whole manuscript, and tried to improve both technically and in the aspect of English language and grammar.
  3. I do agree with the reviewer that the manuscript had many mistakes. In this re-revised manuscript, I tried my best to correct those errors. The changes are highlighted in yellow, and in red font.
  4. In line 240, "the cell surface CD markers are important to measure the extend of COVID-19" has been replaced by "leukocyte subpopulations seem to offer information about the severity of COVID-19"as per reviewer’s suggestion.
  5. Lines 238-244: Here I have re-written the sentences in appropriate way according to reviewer’s suggestion. In addition, I have added the following new sentence (lines 244-247) in order to clarify the role of lymphocytes in COVID-19 severity: “Moreover, Th cells (which help B cells secrete the NAbs and modulate the immune response of other T cells) release IL-2, IFN-γ, IL-4 and other cytokines to activate macrophages and the NK cells [46].”
  6. I have corrected the sentences in lines 266, 271 (shown in red font) and 272 (highlighted in yellow).
  7. I have made correction in line 345.
  8. Regarding the sentences written in lines 352-356, I compiled the experimental data from the associated groups. However, as this paragraph is not clear enough to understand, I have deleted these sentences. I also felt that these sentences are not essential at this stage.
  9. According to the reviewer’s suggestion, I have added a new title (new section 4.2). Also, I have tried to improve English language as well.